# Tibolone Pre-Treatment Ameliorates the Dysregulation of Protein Translation and Transport Generated by Palmitic Acid-Induced Lipotoxicity in Human Astrocytes: A Label-Free MS-Based Proteomics and Network Analysis

**DOI:** 10.3390/ijms23126454

**Published:** 2022-06-09

**Authors:** Diego Julián Vesga-Jiménez, Cynthia A. Martín-Jiménez, Adriana Grismaldo Rodríguez, Andrés Felipe Aristizábal-Pachón, Andrés Pinzón, George E. Barreto, David Ramírez, Janneth González

**Affiliations:** 1Division of Neuropharmacology and Neurologic Diseases, Yerkes National Primate Research Center, Atlanta, GA 30329, USA; vesgad@javeriana.edu.co (D.J.V.-J.); cmart51@emory.edu (C.A.M.-J.); 2Departamento de Nutrición y Bioquímica, Facultad de Ciencias, Pontificia Universidad Javeriana, Bogota 110231, Colombia; mgrismaldo@javeriana.edu.co (A.G.R.); andres_aristizabal@javeriana.edu.co (A.F.A.-P.); 3Laboratorio de Bioinformática y Biología de Sistemas, Universidad Nacional de Colombia, Bogota 111321, Colombia; ampinzonv@unal.edu.co; 4Department of Biological Sciences, University of Limerick, V94 T9PX Limerick, Ireland; george.barreto@ul.ie; 5Departamento de Farmacología, Facultad de Ciencias Biológicas, Universidad de Concepción, Concepcion 4030000, Chile; dramirezs@udec.cl

**Keywords:** mass spectrometry, human astrocytes, tibolone, palmitic acid, obesity, neuroprotection, proteomics, network analysis

## Abstract

Excessive accumulation and release of fatty acids (FAs) in adipose and non-adipose tissue are characteristic of obesity and are associated with the leading causes of death worldwide. Chronic exposure to high concentrations of FAs such as palmitic acid (pal) is a risk factor for developing different neurodegenerative diseases (NDs) through several mechanisms. In the brain, astrocytic dysregulation plays an essential role in detrimental processes like metabolic inflammatory state, oxidative stress, endoplasmic reticulum stress, and autophagy impairment. Evidence shows that tibolone, a synthetic steroid, induces neuroprotective effects, but its molecular mechanisms upon exposure to pal remain largely unknown. Due to the capacity of identifying changes in the whole data-set of proteins and their interaction allowing a deeper understanding, we used a proteomic approach on normal human astrocytes under supraphysiological levels of pal as a model to induce cytotoxicity, finding changes of expression in proteins related to translation, transport, autophagy, and apoptosis. Additionally, tibolone pre-treatment showed protective effects by restoring those same pal-altered processes and increasing the expression of proteins from cell survival processes. Interestingly, ARF3 and IPO7 were identified as relevant proteins, presenting a high weight in the protein-protein interaction network and significant differences in expression levels. These proteins are related to transport and translation processes, and their expression was restored by tibolone. This work suggests that the damage caused by pal in astrocytes simultaneously involves different mechanisms that the tibolone can partially revert, making tibolone interesting for further research to understand how to modulate these damages.

## 1. Introduction

Obesity is defined as an excessive accumulation and release of fatty acids (FAs) in adipose and non-adipose tissue, associated with poor health outcomes [1]. The percentage of people with obesity in the world is growing fast, and this condition is responsible for at least 2.8 million deaths each year. Notably, obesity is a significant risk factor for different chronic diseases [1]. Furthermore, it is known that long-term exposure to circulating high-fats in the body is related to pathological mechanisms in the brain, including Alzheimer’s (AD), Parkinson’s (PD), and Huntington’s Disease [2,3,4,5,6,7,8,9]. One of the most contributive factors to developing an ND is the dysregulation of glial cells, being astrocytes one of the most important players in these pathologies [10]. Among other functions, astrocytes are essential for CNS homeostasis by modulation of synaptogenesis and synaptic transmission, maintaining the blood-brain barrier, neurotransmitter recycling, and ionic balance [11].

In that sense, pal, as the most common saturated fatty acid in the human body [12], can activate different damaging responses in astrocytes, such as inflammation [2], de novo ceramide synthesis [13], and endoplasmic reticulum stress [14]. Furthermore, pal can activate toll-like receptors (TLRs) that trigger a signaling cascade mediated by the nuclear factor enhancing the kappa light chains of B cells (NF-κB) [8,15]. Following nuclear activation and translocation, NF-κB can induce the production of inflammatory cytokines, such as tumor necrosis factor (TNF), interleukin (IL)-6, and IL-1 [16]. In addition, the alteration of pal homeostatic balance in adipose and non-adipose tissues triggers lipotoxic damage [15]. Moreover, pal has a significant effect on mitochondrial functionality since this saturated fatty acid can dampen mitochondrial membrane potential [17,18]. Experimental findings identify these pro-inflammatory pathways and astrocytic mitochondrial dysfunction as the main contributors to different NDs, as mentioned above [11,19]. However, the lack of effective treatments for these conditions highlights the importance of searching for alternatives to improve the patients’ outcomes.

Different works have demonstrated that the neuroprotective and neurotrophic actions of estrogens (as 17β-estradiol) and their cognate receptors can influence the function and structure of the central nervous system [20,21]. However, the potential health risks associated with prolonged exposure to estrogens, such as increased incidence of uterine and breast cancer, may prevent its long-term use [22,23]. Therefore, multiple studies have used estrogen-like compounds as alternatives to obtain therapeutic agents as potential treatments for different NDs [23]. Among these, tibolone, a synthetic steroid, has attracted attention because it shows beneficial effects in acute and chronic NDs [24,25,26,27], and as hormonal therapy for postmenopausal women. Tibolone may exert estrogenic actions in the brain while inducing progestogenic and androgenic effects in the uterus, endometrium, and breast due to its capacity of being metabolized in different compounds generating a specific response in different tissues [26,28,29,30]. It is known that astrocytes express estrogen receptors [30,31,32], which is a target for the protective effects of tibolone [28,33]. Besides, different protective effects of tibolone have been reported on astrocytes, showing it reduces cell death, cardiolipin loss, and nuclear fragmentation after an insult with glucose deprivation [24], and also it has been reported that tibolone reduces cell death and mitochondrial damage against pal [17,28,33,34,35,36,37], and reduces inflammatory response generated by pal on astrocytes [37]. However, the underlying actions of tibolone upon lipotoxic damage are largely unknown.

To address this research gap, and as we proposed in a review pal can induce several mechanisms of damage but most of the studies focus only on specific pathways [38], therefore we used a proteomics approach because it allows a deeper understanding of the processes involved in response [39]. Also, the use of protein-protein interaction combined with systems biology allows an understanding of the impact that those changes in expression will have in a biological system and which proteins can be possible targets for biomarkers [40]. We performed MS-based proteomics and network analysis to assess the effects of supraphysiological levels of pal over normal human astrocytes, as well as tibolone administration effects before the pal challenge. Our results suggest that the main changes induced by pal are related to the translation and transport of proteins, autophagy, and apoptosis. Also, tibolone restored the expression of some of those transport, translation, and apoptosis-related proteins and augmented proteins related to cell survival processes.

## 2. Results

### 2.1. Proteomic-Wide Profiling of Palmitic Acid-Exposed Astrocytes

Our differential expression analysis identified 10,718 peptide groups, corresponding to 1655 proteins (FDR < 0.01). Due to the stochastic nature of “shotgun” label-free or label-free quantitative proteomics, protein identification or abundance data may be missing in certain samples [41]. Therefore, proteins with missing data in any sample after using KNN were excluded from this analysis, resulting in the final quantification of 1281 proteins with complete data in all 18 samples.

After normalization, it was observed that the abundance of proteins was similar between the different conditions and samples used in the analysis (Figure 1A). The abundance of proteins per sample acts as quality control, ensuring the amount of protein in each sample is equivalent before loading onto the mass spectrometer. Furthermore, the principal component analysis (PCA) illustrated general reproducibility and heterogeneity of the different protein expression profiles, grouping the closest samples by treatment and biological replicate. Interestingly, pal 2 mM induced the highest variability in protein expression profiles among the samples (Figure 1B).

Among the 1655 proteins identified, 1115 were commonly identified in all the samples from the three biological replicates without significant differences. Possibly, these proteins compose the core proteins needed for conserving astrocytic function. Nevertheless, this study focuses only on the proteins that showed a significant difference. In each comparison, lists of up-regulated and down-regulated proteins were generated (Appendix A). In general, pal vs. veh presented 54 differentially expressed proteins, tip vs. veh 128, and tip vs. pal 45. Additionally, a comparison between pal vs. veh and tip vs. veh was made, finding 31 shared proteins.

### 2.2. Pal Reduces the Expression of Proteins Related to Transport, Transcription, and Translation Processes

Comparing the normalized abundance of the expressed proteins in pal vs. veh (Appendix A), the obtained PCA presented a grouping by treatment despite the sample heterogeneity (Figure 2A). Interestingly, pal 2 mM induced more variability among the samples. Using these data, 25 up-regulated and 29 down-regulated proteins were obtained (Figure 2B). In addition, these proteins showed a very differentiated pattern in agreement with the treatments and displayed in two clusters according to their expression levels (Figure 2C). The up-regulated proteins showed enrichment for the following terms: in GO Cellular component mainly to the extracellular space; in the GO Biological process Sub-Ontology to the metabolic process of very-long-chain FAs; in KEGG to the FAs metabolism (p_adj = 1.49 × 10^−2^) and biosynthesis of unsaturated FAs (p_adj = 1.55 × 10^−3^) (Figure 2D). Meanwhile, the down-regulated proteins presented enrichment for the following terms: intracellular transport (p_adj = 2.11 × 10^−2^), establishment of localization in the cell, cell localization, initiation of translation (p_adj = 4.33 × 10^−2^), transport, processing of proteins in ER (p_adj = 3.62 × 10^−2^), COPI-mediated transport, cellular response to stress, etcetera (Figure 2E). These results suggest that pal is downregulating the transport of proteins in the cell to different regions, particularly the involvement in ER and up-regulating terms linked to lipidic metabolism.

Interestingly, it was found that pal 2 mM down-regulated different proteins related to translation, such as the ribosomal protein 60S L37 (RPL37, *p* = 2.44 × 10^−4^ and LFC = −1.955268045), a protein that belongs to a segment of the 60S ribosomal subunit. Also, pal negatively regulated Importin-7 (IPO7, *p* = 6.47 × 10^−4^ and LFC = −1.4343906), a protein responsible for transporting proteins to the nucleus [42], such as the ribosomal proteins RPL23A, RPS7, and RPL5 [43], and histones H2A, H2B, H3, and H4 [44]. Furthermore, the mRNA export and transcription factor (ENY2, *p* = 3.82 × 10^−4^ and LFC = −1.696939985) was negatively regulated too. This protein is involved in the activation of transcription-coupled to mRNA export [45], and the reduction of transcription can be linked with the decrease of the translation process [46]. Another down-regulated protein was the eukaryotic translation initiation factor 4 gamma 2 (eIF4G2, *p* = 2.61 × 10^−3^ and LFC = −2.91627833), a protein related to the initiation of translation and inhibition of local protein synthesis and axon growth [47].

In addition to reducing the translation initiation pal increased processes related to the biosynthesis of FAs and their transformation to more complex molecules. Among the proteins that pal positively regulated, we found the 3-ketoacyl-CoA peroxisomal thiolase EC 2.3.1.16 (ACAA1, *p* = 7.49 × 10^−5^ and LFC = 2.60276577); very-long-chain enoyl-CoA reductase (TECR, *p* = 5.39 × 10^−5^ and LFC = 1.89963979), and very-long-chain Fas elongation protein 1 (ELOVL1, *p* = 6.39 × 10^−3^ and LFC = 1.55414505), which enrich the metabolism of very-long-chain Fas (p_adj = 1.63 × 10^−2^) and for the metabolism of Fas (p_adj = 1.49 × 10^−2^).

### 2.3. Tibolone Regulates the Expression of Proteins Related to Translation, Transport, and Immune Response

In the tip vs. veh comparison (Appendix A), samples showed a clear separation according to the treatment, being NHA3 veh the only outlier (Figure 3A). In this case, we found 57 up-regulated proteins and 71 down-regulated, as seen in the obtained volcano plot (Figure 3B). The heatmap in Figure 3C shows the strong differentiation between the expressed proteins in tip vs. veh. Here we can appreciate the proteins that make NHA3_veh2 different from the other veh samples (right corner). The up-regulated proteins by the pre-treatment with tibolone 10 nM showed enrichment for terms related to regulation of organelle organization (p_adj = 1.45 × 10^−2^), extracellular exosome (p_adj = 3.81 × 10^−8^), and neutrophil degranulation (p_adj = 2.73 × 10^−3^) (Figure 3D). On the other hand, tibolone decreased proteins related to metabolism of mRNA (p_adj = 1.40 × 10^−2^), GTP hydrolysis of 60S ribosomal unit (p_adj = 1.12 × 10^−2^), translation (p_adj = 6.59 × 10^−3^), and catabolic process of macromolecules (p_adj = 3.46 × 10^−2^) (Figure 3E). Therefore, at the enrichment level, the protective effect of tibolone is probably related to the negative regulation of processes that affect protein synthesis and counteract the dysregulation in the transport and localization of proteins. In addition, tibolone pre-treatment regulates proteins associated with the immune response.

Additionally, we identified 17 up-regulated and 28 down-regulated proteins in tip vs. pal (Figure 4A, Appendix A). In a heatmap, those differentially expressed proteins showed a congruent separation between treatments (Figure 4B). The terms for up-regulated proteins were related to the COPII vesicle coat (p_adj = 8.89 × 10^−3^) and ER exit site (p_adj = 3.91 × 10^−2^) (Figure 4C). The enriched terms for the down-regulated proteins were extracellular organelle (p_adj = 7.48 × 10^−3^), extracellular vesicle, extracellular exosome (p_adj = 6.58 × 10^−3^), and cadherin binding (p_adj = 3.02 × 10^−2^) (Figure 4D). It suggests that the main differences between tip and pal are related to transport and vesicles.

### 2.4. Tibolone Restores Expression Levels in Transport, Translation, and Apoptosis-Related Proteins

Differentially expressed proteins were compared in the case of pal vs. veh and tip vs. veh. Worth of note, 31 proteins were found regulated in both comparisons (Table 1). These proteins presented the same expression pattern, except for SRP68 or t“e“ signal recognition particle subunit SR”68” that changed from a down-regulation in the pal vs. veh to up-regulation in tip vs. veh (Figure 5A). Notably, when observing the shared proteins between the tip vs. veh and pal vs. veh comparisons, it is found that the up-regulated proteins show enrichment for the synthesis of unsaturated FAs such as alpha-linoleic acid, suggesting that their synthesis is augmented in both treatments that involve PA (Figure 5B). Among the down-regulated proteins, we found an enrichment in cytoplasmic dynein complex (p_adj = 2.90 × 10^−2^), response to heat (p_adj = 3.25 × 10^−3^), and protein processing in RE (p_adj = 9.74 × 10^−3^) (Figure 5C).

Alternatively, pal vs. veh compared with tip vs. veh showed 13 up-regulated unique proteins and 10 down-regulated, suggesting that tibolone restored the expression level on those 23 proteins (Figure 5A, Appendix A). Among them, the down-regulated IPO7, whose role is related to protein transport. According to the results of GOSlim (Appendix A), the down-regulation of protein transport affects the transport of peptides, amines, macromolecules, and the establishment of proteins in the compartments. Therefore, it would affect the functioning of the ER and the possible destinations of these vesicles if tibolone reverts the expression of these proteins to vehicle-like expression. Furthermore, the related to the translation RPL37 and eIF4G2 protein expression were also restored, suggesting the protective effect of tibolone on human astrocytes.

### 2.5. ARF3 and IPO7 as Key Proteins in the Lipotoxic Scenario Triggered by Pal

A proteomic signature linked to the different treatments was obtained from our unsupervised hierarchical clustering analysis, using protein abundances in the 18 samples (Figure 6). A total of 11 protein modules were generated (blue, yellow, turquoise, pink, red, green, magenta, purple, black, brown, and grey) (Figure 6A,B). Interestingly, most of the proteins were grouped according to the module-trait association, the red module had the highest correlation coefficient followed by turquoise and pink (Figure 6C). Thus, the three modules were selected for further analysis. Notably, GS versus MM reflected a strong relationship between treatment and genes in these modules, implying that hub genes of these modules also tend to be highly correlated with the treatments (Figure 6D). Additionally, the results revealed that the red, pink, and turquoise modules were the more correlated with the weight trait, which could be interpreted as a demonstration of the essentially of these modules in the responses generated by treatment (Figure 6F).

The proteins from the selected modules generated 355 nodes and 2657 edges, which were reduced to 110 hub proteins and 856 edges after the MCODE algorithm (Figure 7 and Appendix A). Cluster 1, including 42 nodes, presented an MCODE score of 35,220 (Figure 7A), while cluster 2, with 17 nodes, had a score of 7375 (Figure 7B). Clusters 3 to 7 with MCODE score > 3500 presented 8 to 4 nodes (Figure 7C), and clusters 8 to 14 with MCODE score > 3000 displayed 4 to 3 nodes (Figure 7D). These proteins, having a higher weight in the network with various interactions, are relevant in the system and may be possible targets for understanding lipotoxic damage and tibolone treatment.

To define the key proteins, the proteins present in the hubs were intercepted by those differentially expressed, with an adjusted *p*-value or *q*-value < 0.05. We obtained 27 proteins that fulfilled this condition, of which 10 are related to the response induced by the lipotoxic damage generated by pal 2 mM and 17 to the action of tibolone in human astrocytes (Table 2). The up-regulated ARF3 and the down-regulated IPO7 proteins remained exclusive from pal vs. veh (Table 2). Our unsupervised hierarchical clustering analysis included these two proteins in cluster 6 with MCODE score = 4000 (Figure 7C in yellow). According to the Open Target Platform, ARF3 and IPO7 are related to different diseases. Focusing on the nervous system disorders, ARF3 (Figure 8A) presented a higher association score, and it was related to more diseases than IPO7, which was more related to measurement, cancer, and gastrointestinal diseases (Figure 8B).

### 2.6. Experimental Validation

To validate our proteomic data, IPO7 was selected based on its fold change and high confidence (p_adj = 3.32 × 10^−2^ and LFC = −1.4343906). Additionally, IPO7 was identified as a key protein in this study, making it a good candidate. Figure 8 shows how pal reduced the IPO7 expression with a mean of 12.62% compared with veh (*p* = 3.60 × 10^−2^). On the other hand, pal against tip presented no significant difference (*p* = 0.357), neither did veh vs. tip (*p* = 0.3608). These results are aligned with the findings of our proteomics data.

## 3. Discussion

Astrocytes have a fundamental role in brain homeostasis, and the understanding of how they react upon lipotoxicity would improve our current knowledge about ND progression [48,49,50,51]. This study presented an elaborate proteome analysis of human astrocytes under pal lipotoxic conditions and compared these results with the obtained protein expression during a protective treatment using tibolone. We identified 1655 proteins, of which 54 were differentially expressed in pal compared with the control, 128 were regulated in the tibolone 10 nM pre-treatment prior to pal, and 45 in the comparison between the two different treatments. The number of identified non-significant or significant differentiated proteins agrees with other works using human astrocytes [52,53,54].

The pal effects on the cell over the protein translation process have been barely studied, with none to date at the nervous system level. However, one study in macrophages suggests that pal 2 mM reduces translation [7]. Similar to previous works, the present study results indicate that pal induces the expression of the eukaryotic initiation factor 2α (eIF2α, *p* = 5.33× 10^−3^ and LFC = −1.16389530), which represses the translation process [8,55]. Nevertheless, recent findings suggest that expression and phosphorylation of eIF2α do not necessarily lead to down-regulation of global translation, and no mandatory connection should be assumed [55]. Therefore, the role of eIF2α dysregulation must be validated independently and not solely by its expression/phosphorylation. On the other hand, it is interesting that mutations in the enzymes responsible for phosphorylating eIF2α lead to human diseases, which often include neurological and neurodegenerative pathologies [56].

Besides, our proteomic analysis showed that pal 2 mM negatively regulated the expression of subunit 1 of eukaryotic translation initiation factor 2 (eIF2S1, *p* = 4.18 × 10^−3^ and LFC = −1.14091479). This protein acts in the first steps of protein synthesis, forming a ternary complex with GTP and as an initiator of RNA translation [57]. However, it also has the dual ability to repress protein translation [7]. Also, the mRNA export and transcription factor (ENY2, *p* = 3.81 × 10^−4^ and LFC = −1.696939985) was negatively regulated. This protein is involved in the activation of transcription-coupled to mRNA export [45], and a reduction in transcription can reduce translation [46].

The down-regulation of proteins related to transcription and translation suggests changes in the mechanism of protein translation, with dysregulation of factors such as eIF4 and eIF2α and other proteins related to the translation initiation process such as RPL37 and IPO7 that regulate other RPLs [8,58]. Additionally, it has been reported that in AD, the protein translation machinery is altered, which is mainly seen in elongation factors and ribosomal proteins [58]. At the cellular level in the AD model, astrocytes showed a reduction in ribosomal binding and translation-related proteins [59]. Similarly, reduced protein synthesis has been associated with PD progression [60]. That indicates that the lipotoxicity generated by pal could be related to the translation impairment seen in the progression of some NDs [60,61].

After pal treatment, ubiquitin-protein ligase E3 CHIP was strongly down-regulated by pal (STUB1, *p* = 1.91 × 10^−4^ and LFC = −2.805049685). This protein has a fundamental role in the protein folding process directed at misfolded chaperone substrates toward proteasomal degradation, and it probably induces polyubiquitination, necessary for protein degradation [61]. Besides, pal down-regulated the expression of the ubiquitin recognition factor in the ER 1-associated degradation protein (UFD1, *p* = 1.99 × 10^−4^ and LFC = −1.369768858) which is a key component for the ubiquitin-dependent proteolytic pathway for the degradation of misfolded proteins [42]. In addition, pal up-regulates pro-apoptotic proteins such as programmed cell death protein 5 (PDCD5, *p* = 1.42 × 10^−3^ and LFC = 1.864688494) and interferon-induced protein with tetratricopeptide repeats 2 (IFIT2, *p* = 3.63 × 10^−3^ and LFC = 2.561125172). Up-regulation of PDCD5 and IFIT2, combined with the down-regulation of STUB1 and UFD1, suggest a mechanism by which exposure to pal disrupt autophagy and apoptosis induction. It has been reported that pal induces cell death in astrocytes [17,37,62], reduction of the autophagy process, and activation of pro-apoptotic pathways [16,63]. Surprisingly, according to our results, pal-induced damage in human astrocytes was not strongly related to oxidative stress or inflammatory response at the protein level.

The results of this study additionally show that among the unique pal proteins up-regulated are TECR (*p* = 5.39× 10^−5^ and LFC = 1.89963979), SUCLA2 (*p* = 5.66 × 10^−3^ and LFC = 0.94695906), and ELOVL1 (*p* = 6.39 × 10^−3^ and LFC = 1.55414505), enriching the FA metabolism and FA elongation pathways. That suggests a relationship between the synthesis of very-long-chain FAs and the synthesis of sphingolipids through the metabolic pathway of sphingosine 1-phosphate (S1P) [63]. Furthermore, the accumulation of very-long-chain FAs in the brain is related to demyelination caused by peroxisomal pathologies [64]. On the other hand, the dysfunction of the S1P receptor signaling system increases several vascular defects, such as angiogenesis and increased inflammation, due to the increased permeability it generates in the blood vessels [65]. Although, it should not be stated that the effect of S1P is detrimental since it can have a dual role in the brain, resulting in protection in some conditions and harmful in others [19,66]. Therefore, it would be interesting to study further the S1P effects on human astrocytes under lipotoxic damage with pal.

Another up-regulated protein only in pal is RuvB-like 1 (RUVBL1, *p* = 1.94 × 10^−3^ and LFC = 1.13983237). This protein is responsible for acetylating histones H4 and H2A, activating the transcription of different genes [66]. Some of these are associated with oncogenes, apoptosis, senescence, or DNA repair [66]. Interestingly, this protein is also related to cell proliferation [67]. Considering that TECR, SUCLA2, ELOVL1, and RUVBL1 are unique proteins in the pal treatment, it would mean that tibolone managed to restore their expression levels. That is consistent with the observed in preliminary reports showing that part of the protective effect of tibolone on astrocytes is related to the regulation of inflammation [28,68].

Compared to the vehicle, tibolone pre-treatment increased the transport protein Sec24C (Sec24C, *p* = 2.93 × 10^−3^ and LFC = 1.576720125) and the programmed cell death protein 6 (PDCD6, *p* = 2.01 × 10^−3^ and LFC = 1.339618944) as well. The two proteins are related to the formation of the COPII vesicle coating and ER output, indicating that they favor cellular transport, avoiding the dysregulation of protein localization [68]. PDCD6 plays different cell function roles as the regulation of cell proliferation and vesicular transport from ER to Golgi [69]. Also, PDCD6 is involved in membrane repair, stabilization of weak protein interactions [70,71], and participates in the acceleration of apoptosis by increasing caspase 3 activity [71]. However, in previously obtained experimental data, tibolone 10 nM pre-treatment reduced cell death in NHA treated with pal [36], and therefore, further studies are needed to determine which role this protein is playing.

Tip reduced lysosomal alpha-glucosidase (GAA, *p* = 1.60 × 10^−4^ and FDR = −1.701963592), suggesting that tibolone pre-treatment reduces glycogen catabolic processes related to ROS production [72,73]. Alternatively, it was found that tip reduces the expression of cytoplasmic aconitase (ACO1, *p* = 1.41 × 10^−5^ and FDR = −2.688103819). This protein is responsible for regulating iron homeostasis, acting as a chelator [73], and its reduction could be harmful to cells. These results are aligned with the findings of a study using hormone replacement therapy with tibolone, showing that tibolone use alters the expression of proteins linked to energy obtention and metabolism [74]. Also, these results suggest that tibolone has some protective effects on NHA cells against the effects of pal but does not entirely reverse them.

In the unique proteins observed in tip vs. veh, IFIT3 was up-regulated (*p* = 1.34 × 10^−3^ and LFC = 2.054834577). This protein acts as an inhibitor of cellular and viral processes, cell migration, viral proliferation, signaling, and replication [75]. IFIT3 increases CDKN1A/p21 and CDKN1B/p27 expression, negatively regulating the cell cycle. Classically, the turnover of CDKN1B/p27 is regulated by COPS5, which binds CDKN1B/p27 in the nucleus and exports it to the cytoplasm for its ubiquitin-dependent degradation [76]. IFIT3 sequesters COPS5 in the cytoplasm, increasing the levels of the nuclear protein CDKN1B/p27. In addition, it up-regulates CDKN1A/p21 by down-regulating MYC, a repressor of CDKN1A/p21 [76]. Furthermore, this protein can negatively regulate the pro-apoptotic effects of IFIT2 [77], a protein increased by pal.

In addition, it was also found that mitogen-activated protein kinase 4 (MAP4K4, *p* = 1.87 × 10^−3^ and LFC = 0.93110042) may play a role in response to environmental stress and cytokines such as TNF-α [78]. Moreover, this protein plays a role in the induction of ARF transduction [79] and the negative regulation of apoptosis [80]. Therefore, MAP4K4 and IFIT3 could be contributing to attenuate the apoptotic processes generated by pal 2 mM, and IFIT3, in addition, could be preventing excessive proliferation, which is part of the typical astrogliosis process [32,33,62].

The protective response of tibolone could also be explained by the differentially expressed proteins in the tip vs. pal comparison. One of these proteins is the carboxyterminal ubiquitin hydrolase 14 (USP14, *p* = 6.99 × 10^−4^ and LFC = −1.89683838), which acts as a physiological inhibitor of ER-associated degradation through interaction with ERN1. According to that, tibolone could be altering the regulation of autophagy generated by pal in astrocytes [14].

When comparing tip against pal, the down-regulated drebrin-like protein (DBNL, *p* = 2.77 × 10^−5^ and LFC = −3.104115258) is additionally observed. DBNL binds to actin and plays a role in its polymerization [42]. However, DBNL also activates the N-terminal c-Jun kinase (JNK) [81]. This process is related to pro-apoptotic signaling [82], which indicates that tibolone has a role in its regulation, contributing to the protective response observed in previous studies [36]. In addition to that, DBNL is also associated with neutrophil degranulation, which are related to inflammatory processes [83].

Focusing on the proteins shared between pal vs. veh and tip vs. veh, the SRP68 protein stands out, which is necessary to translocate proteins to the ER [84]. This protein went from down-regulated in pal vs. veh to up-regulated in tip vs. veh. It has been observed that when generating a reconstituted SRP in the absence of the SRP68-SRP72 heterodimer, it lacks the elongation and translocation arrest activity [85,86]. The elongation arrest function is physiologically important in mammalian cells since the efficiency of protein translocation to the ER is significantly reduced when the SRP elongation arrest function is canceled, affecting its function [87,88].

Furthermore, the proteins that presented the higher weight in the network and significant differences in expression levels were considered key proteins. When they are modified, it can seriously disturb the system due to the high number of interactions they have [88]. Therefore, exclusively differential expression of proteins in pal with a high weight in the networks highlighted two key proteins: ARF3 up-regulated and IPO7 down-regulated. IPO7 is related to the activation of p53, a characteristic of ribosomal biogenesis stress and increased binding of Mdm2 to ribosomal proteins L5 and L11 (RPL5 and RPL11) [89]. Besides, IPO7 depletion affects the transport of other ribosomal proteins such as RPL23A, RPS7, and RPL5, affecting ribosomal biogenesis [90,91]. It has been reported that RPL5 and RPL11 induce protein translation inhibition and blockage in the cell cycle G1 and G2/M phases through p53 induction [91]. Interestingly, RPL37 (*p* = 2.44 × 10^−4^ and LFC = −1.955268045) was negatively regulated by pal 2 mM, reinforcing the pal’s possible involvement in the protein translation machinery [92]. Of particular interest, IPO7 and RPL37 showed no significant differences between tip and veh. It could mean that the tibolone protective effect would have a role in the expression of these proteins restoring the vehicle levels. This observation agrees with the findings in a tauopathy mouse model, showing a reduction in the synthesis of ribosomal proteins such as some RPS and RPL [93].

On the other hand, the ARF3 is related to Golgi transport toward different regions [42], specifically with retrograde transport to the ER [94]. Thus, pal may modify the translation of the proteins and their transport. Additionally, pal is found to reduce the expression of dynactin subunit 2 (DCTN2, *p* = 8.59 × 10^−6^ and LFC = −2.14658304). This subunit is related to the transport from the ER to the Golgi, and the formation of the mitotic spindle, interacting directly with the other subunits of actin [95,96]. The above has also been reported on gene and protein functions in NCBI (DCTN2 dynactin subunit 2 [Homo sapiens (human)]—Gene—NCBI). That is interesting because, in a mouse model, it was observed that dysfunction of the dynein/dynactin ratio led to the development of amyotrophic lateral sclerosis [96]. Additionally, it has been reported that the overexpression of DCTN2 is harmful in the axonal transport of motor neurons and leads to the development of neuromotor diseases [97,98]. However, in this study, DCTN2 is down-regulated, and the possible effects of its down-regulation in the brain are not clear.

Our proteomic data did not find differentially expressed proteins directly related to an inflammatory response. However, many pal proteins are related to the term neutrophil degranulation, and this could indirectly indicate that pal is causing inflammation through the liberation of pro-inflammatory cytokines and ROS [98]. Furthermore, this study showed that pal up-regulates the proteins related to the transformation of pal to other more complex intermediates through TECR, ELOVL1, and ACAA1, which are linked to FA metabolism and the elongation of very-long-chain FAs. This result is interesting because when pal accumulates in the cell, it can be transformed into diacylglycerol and ceramides, and these can activate several signaling pathways common for lipopolysaccharide-mediated activation of TLR4 [7]. Besides, pal metabolic products are known for modulating the activation of various protein kinase catalytic subunits (PKCs), ER stress and increased ROS generation, and even inflammation and cell death [99,100].

The damage caused by pal has been reported to trigger an inflammatory response in astrocytes and is associated with harmful mechanisms such as oxidative stress, ER stress, and autophagic defects [16,63]. In this sense, it has also been reported that tibolone exerts protective functions against inflammation in experimental neuronal models [27]. Furthermore, the findings of this study agree with literature reports with astrocytes treated with 1 mM pal for 24 h with no ROS production [101]. That could be related to mechanisms of damage and cell death generated by pal independent of ROS production [100], probably since pal is not being used for β-oxidation pathway, but increasing ceramide synthesis [14,102]. Finally, it is essential to remember that tibolone must be metabolized, and the protective effect in the brain is generated by 3-alpha-hydroxy- and 3-beta-hydroxy-tibolone [27], leading to its correct metabolism dependence to present the mentioned protective effects, and this could explain the ambiguous effects presented by tibolone pre-treatment.

## 4. Materials and Methods

### 4.1. Cell Culture

Normal Human Astrocytes (NHA, Lonza, Basel, Switzerland, CC-2565) were used for this study due to their inherent similarity in morphology and function to primary astrocytes. In addition, NHA cells do express GFAP (Glial Fibrillary Acid Protein), a key marker of astrocytes. Three batches of NHA cells (#0000612736, #00005656712, #0000514417) were cultured in ABM (Astrocytes basal medium, Lonza) supplemented with AGM (Astrocyte Growth Medium BulletKit™ Lonza, Basel, Switzerland, Catalog #: CC-3186). The three batches were trypsinized in the second passage and placed in flasks of 25 cm^2^ at a density of 10.000 cells/cm^2^. The cells were incubated at 37 °C and 5% CO_2_ for 12 days in ABM. Until the cells reached a confluence near 80%, the medium was changed every two days, as recommended by the providers. Then cells were submitted to 6 h of serum deprivation and following this the three different treatments were applied: the first one is the vehicle as a control containing 1.35% BSA (fatty acid-free bovine serum albumin; Sigma A2153)+ 2 mM of carnitine and DMSO 0.000025%; the second one is pal 2 mM for 24 h; the third one is tibolone 10 nM pre-treatment for 24 h before pal 2 mM and, obtaining three biological replicates and two technical replicates for each treatment, for a total of 18 samples.

### 4.2. Tibolone Pre-Treatment

Cells were treated with tibolone before the addition of pal. Tibolone (Lot T0827, Sigma, St Louis, MO, USA) was dissolved in 100% DMSO as a stock solution at 40 mM, and further dilutions were made with serum-free DMEM up to a final concentration of 0.000025%. The aliquots were stored at −20 °C, and each aliquot was used three times or less. Varying times and concentrations of tibolone treatment were tested, and the pre-treatment of tibolone 10 nM for 24 h was found to best preserve cell viability upon pal treatment [36].

### 4.3. Palmitic Acid Treatment

NHA cells were washed with 1X PBS and then treated with serum-free DMEM containing pal (Sigma), BSA (fatty acid-free bovine serum albumin; Sigma A2153) as a carrier protein, and carnitine (Sigma, St Louis, MO, USA) to transport FAs into the mitochondrial matrix. Cells were treated at different times using distinct concentrations of pal. Previous results from Martin-Jiménez, (2020) [36] indicated that the optimal pal concentration was 2 mM diluted in 1.35% BSA for 24 h. The control group included 1.35% of BSA and 2 mM carnitine [36].

### 4.4. Protein Extraction and Quantification

Before extraction, the medium was removed, and the flask was washed with 1 mL of cold 1X PBS, which was discarded with a pipette. Lysis buffer for protein extraction was composed of 720 mL RIPA buffer (Thermo Fisher Scientific, Waltham, MA, 174 USA, Catalog number: 89901); 10 mL of 10 mM sodium fluoride; 10 mL of Halt™ Protease Inhibitor Cocktail (100×) (Thermo Fisher Scientific, Waltham, MA, 174 USA, Catalog number: 78430); 250 mL of 2.5 mM pyrophosphate; 10 mL of 1 mM orthovanadate. For a 25 cm^2^ flask, we added 72 mL of the lysis buffer, incubating the flask for 10 min at −4 °C, then the entire surface was scraped and collected in an Eppendorf. At this point, the Eppendorf tubes were placed in ice for 30 min and vortexed for 10 to 15 s every 10 min. Then, samples were centrifuged at 15,200 rpm and −4 °C for 13 min. Finally, the obtained supernatants were transferred to a new tube and stored at −80 °C. The amount of protein per sample was quantified by the Bicinchoninic acid assay (BCA) method with the Pierce ™ BCA Protein Assay Kit (Thermo Fisher Scientific, Waltham, MA, 174 USA, Catalog number: 23227), following the manufacturer’s instructions.

### 4.5. Protein Digestion and Load in the Q Exactive

The protein pellet was then solubilized in 200 µL of 6 M urea and submitted to the UC Davis Proteomics Core. For digestion, 200 mM of dithiothreitol (DTT) was added to a final concentration of 5 mM, and samples were incubated for 30 min at 37 °C. Next, 20 mM iodoacetamide (IAA) was added to a final concentration of 15 mM and incubated for 30 min at room temperature, followed by the addition of 20 µL DTT to quench the IAA. Next, Lys-C was added to the sample and incubated for 2 h at 30 °C. Samples were then diluted to >1 M urea by adding 50 mM AMBIC, and the samples were digested overnight at 37 °C using trypsin. The following day, the samples were desalted using the Macro Spin Column (Nest Group Inc. 17 Hayward St., Ipswich, MA 01938-2041, USA; Catalog number: NC1418588).

Digested peptides were analyzed by LC-MS/MS on a Thermo Scientific Q-Exactive Orbitrap Mass spectrometer along with Proxeon Easy-nLC II HPLC (Thermo Fisher Scientific, Waltham, MA, 174 USA) and Proxeon nanospray source with a spray voltage of 2.0 kV, Capillary temp is 250 c, not gas was used. The digested peptides were loaded with an injection volume of 10 µL onto a 100-micron × 25 mm Magic C18 100 Å 5 U reverse-phase trap where they were desalted online before being separated using a 75-micron × 150 mm Magic C18 200 Å 3 U reverse-phase column. Peptides were eluted using a 180-min gradient with a 300 nL/min flow rate. An MS survey scan was obtained for the *m*/*z* range 300–1600, and MS/MS spectra were acquired using a top 15 method, where the top 15 ions in the MS spectra were subjected to HCD (High Energy Collisional Dissociation). Targeted precursor selection uses an isolation mass window of 2.0 *m*/*z*, and normalized collision energy of 27% was used for fragmentation. A twenty-second duration was used for dynamic exclusion.

### 4.6. Raw Files Processing for Protein Identification

The files were processed using the following parameters: a maximum value of miss cleavage of 2, a minimum of 50% in ion precursor identification, and a search of razor and unique peptides. The database was generated using the validated proteins from SwissProt human. The label-free-quantification was generated with proteome discoverer 2.3 using Sequest and AMANDA search engines. Also, the results were processed using MaxQuant v1.6.10.43 and Perseus v.1.6.10.45 to compare the number of valid proteins and identified peptides. Sequest obtained results with more proteins and better clustering of the samples.

### 4.7. Normalization and Statistics for Relative Quantification

Protein intensities (non-normalized) were imported into the R statistical programming software version 4.0.1 (R Core Team, 2019) for data processing and statistical analysis. Data were Log2-transformed, almost routinely made to obtain a more symmetrical distribution before statistical analysis. All proteins with 70% valid values per group (comprising 6 replicates in each group) were kept [41]. Normalization was performed by the Variance Stabilization Normalization (VSN) method, one of the non-linear methods that aim to maintain constant variance throughout the range of data and approaches the logarithm of large values to eliminate heteroscedasticity using inverse hyperbolic sine [102].

The imputation of missing values was performed by K-nearest Neighbor (KNN), which identifies k features that are very similar to the proteins with missing values, where the similarity is estimated by the Euclidean distance measure, and the missing values are imputed with the values of the weighted average from these neighboring proteins [103,104]. Using a KNN value of 10 as suggested by Välikangas et al. (2017) [103].

### 4.8. Differential Expression Analysis

The differential expression analysis was performed using the Optimized Reproducibility Test (ROTS) statistic, which classifies features according to their expression. The ROTS is a modified *t*-test that eliminates the bias in the data [105,106]. Differential expression was assessed between three pairs of conditions: (1) pal versus control (veh), (2) pal with tibolone pre-treatment (tip) versus veh, and (3) tip versus pal. Subsequently, the q-value was calculated with the methods of pFDR, Benjamini Hochberg using the R package qvalue 2.18.0 Bioconductor (https://www.bioconductor.org/packages/release/bioc/html/qvalue.html, accessed on 15 February 2020) [106]. Differentially expressed proteins were selected by |Log2 FC (LFC)| ≥ 0.58, *p* < 0.01, and FRD < 0.1. Appendix A shows a schematic overview of the sample preparation and MS-based proteomics analysis conducted in this work.

### 4.9. Functional Enrichment Analysis

Protein set enrichment analysis was conducted using g:Profiler (accessed on 15 May 2020) [107]. To search for the TOP 20 terms, *p*-adjusted significance was set at Bonferroni-corrected threshold of *p* < 0.05 of the protein relationship across Gene Ontology terms, giving priority to biological process and molecular function, pathways from KEGG, Reactome [108], and Human Protein Atlas HPA.

### 4.10. Weighted Co-Expression Network Analysis

Weighted gene co-expression network analysis (WGCNA) package in R was used to construct a co-expression network for the protein-protein interactions present in the data set, using the changes in the expression of each protein in the treatments with the normalized values of the protein abundances [109]. A soft threshold power β of 9 (β = 9) and minimum module size of 20 were used to generate the different modules. First, Pearson’s correlation matrices were calculated between pairwise proteins to classify proteins with similar expression profiles into modules; differentially expressed proteins were assessed using Bonferroni-corrected Fisher’s exact test *p*-values. Next, Genetic significance (GS) was used to measure the degree of association between the proteins, and the trait module Membership (MM) was used to determine the location of a global network. Following this, the construction of PPI networks and identification of critical targets or hub proteins was built using the MCODE plugin of Cytoscape 3.8.0 [110]. Finally, we intersected those hub proteins with the proteins differentially expressed with adjusted *p*-value (p_adj) < 0.05 in each comparison to determine proteins with high relevance in the network that changed their expression with high confidence.

### 4.11. Experimental Validation

NHA protein extracts from 5 different replicates were loaded with equal amounts of protein (30 µg), separated by 10% SDS-PAGE, and electro-transferred to a nitrocellulose membrane (Thermo Fisher Scientific, Waltham, MA, 174 USA). Membranes were blocked with 5% BSA (bovine serum albumin) for 2 h at room temperature. After that, the membranes were incubated overnight at 4 °C with anti-IPO7 antibody (1:1000 dilution; Abcam, Cambridge, UK; Catalog number: ab99273), and anti-β-actin antibody (1:50,000 dilution; MilliporeSigma, 400 Summit Dr, Burlington, MA 01803 USA; Catalog number: JLA20), followed by three washes with 1X Tris-Buffered Saline, 0.1% Tween (TBS-T) for 5 min each one. Then, the membranes were incubated with the respective anti-rabbit and anti-mouse secondary antibody (1:10000, Santa Cruz Biotechnology 10410 Finnell Street Dallas, TX 75220, USA) for 1 h. Visualization was performed using enhanced chemiluminescence SuperSignal™ West Pico PLUS Chemiluminescent Substrate (Thermo Fisher Scientific, Waltham, MA, 174, USA Catalog number: 34577) and revealed using the equip iBright 1500 (Invitrogen, Waltham, MA, 174, USA). The expression levels of each protein were quantified by measuring protein intensities on immunoblot images using the Image J software version 1.53i March 2021 (National Institutes of Health, Bethesda, Maryland 20892, USA) and normalized to the corresponding level of β-actin in each sample. The protein abundances were normalized across the samples with the control (veh) and compared using one-way ANOVA and the post hoc Tukey’s test to compare the samples; *p* < 0.05 was considered statistically significant.

## 5. Conclusions

This one is the first comprehensive study of the proteomic profile of human astrocytes subjected to lipotoxic damage and how they respond to a tibolone 10 nM pre-treatment. Our results expand the understanding of the lipotoxic effect of palmitic acid 2 mM on human astrocytes. The weighted gene co-expression network analysis identified 110 hub proteins, and 27 of them were differentially expressed, of which 10 are exclusively related to the response induced by lipotoxic damage generated by pal and 17 to the action of tibolone in human astrocytes. Pal regulated proteins related to activation of immune response, dysregulation of protein synthesis, autophagy, vesicle transport, and ER-related processes. We also found some effects generated by pal, such as reducing some ribosomal proteins and dysregulating protein translation. Interestingly, tibolone returned the expression of several of these proteins to vehicle levels.

Additionally, tibolone reduced the expression of proteins that activate pro-apoptotic pathways. However, the tibolone response showed effects that may seem detrimental to the cell, such as reducing the expression of aconitase and MICOS complex or increasing a protein with a pro-apoptotic activity. Therefore, it cannot be said that tibolone completely reverses the damage caused by pal, but it does show protective effects at the astrocyte level that needs to be studied at the crosstalk among the different cells in the brain. According to our results, it is likely that the beneficial tibolone effects may outweigh the harmful ones, doing it a promising treatment for lipotoxic damage in human astrocytes.

## Figures and Tables

**Figure 1 ijms-23-06454-f001:**
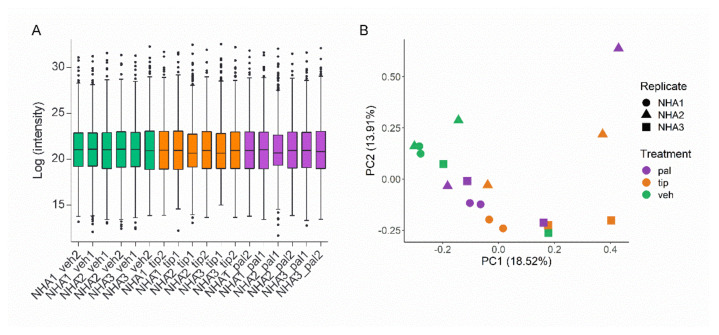
Protein expression profile in normal human astrocytes (NHA) under lipotoxic treatments. Three biological (NHA 1 to 3) and two technical replicates were cultured in medium alone (veh), medium containing palmitic acid (pal), or medium containing tibolone 10 nM pre-treatment and palmitic acid 2 mM (tip). (**A**) Tukey boxplot of Protein abundance by sample, normalized with VSN method. (**B**) Principal component analysis (PCA) to evaluate the relationships between biological replicates and treatments. Green: veh samples, orange: tip samples, and violet: pal samples.

**Figure 2 ijms-23-06454-f002:**
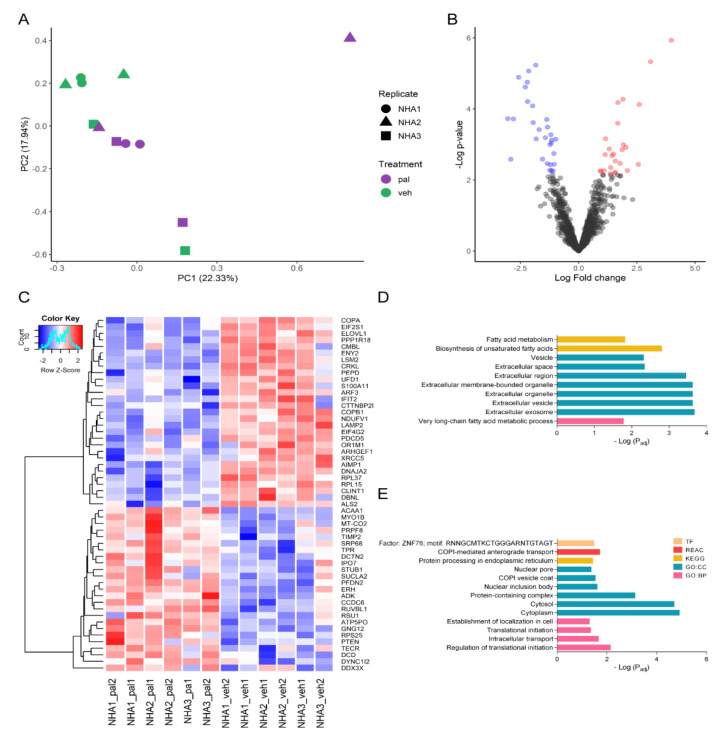
Differential expression of pal vs. veh. (**A**) PCA to evaluate the relationships between biological replicates and pal vs. veh treatments. (**B**) Volcano plot showing differentially expressed proteins with adjusted *p*-value < 0.01 and |Fold change| ≥ 1.5. (**C**) Heatmap of proteins differentially expressed proteins in pal vs. veh. Color scale: variation of normalized protein abundance (Z-Score). (**D**) Enrichment analysis of up-regulated proteins in pal vs. veh (*p* < 0.05). (**E**) Enrichment analysis of down-regulated proteins in pal vs. veh (*p* < 0.05). TF: TRANSFAC database; REAC: Reactome database; KEGG: Kyoto Encyclopedia of Genes and Genomes; GO:CC: Gene Ontology cellular component; GO: BP: Gene Ontology biological process.

**Figure 3 ijms-23-06454-f003:**
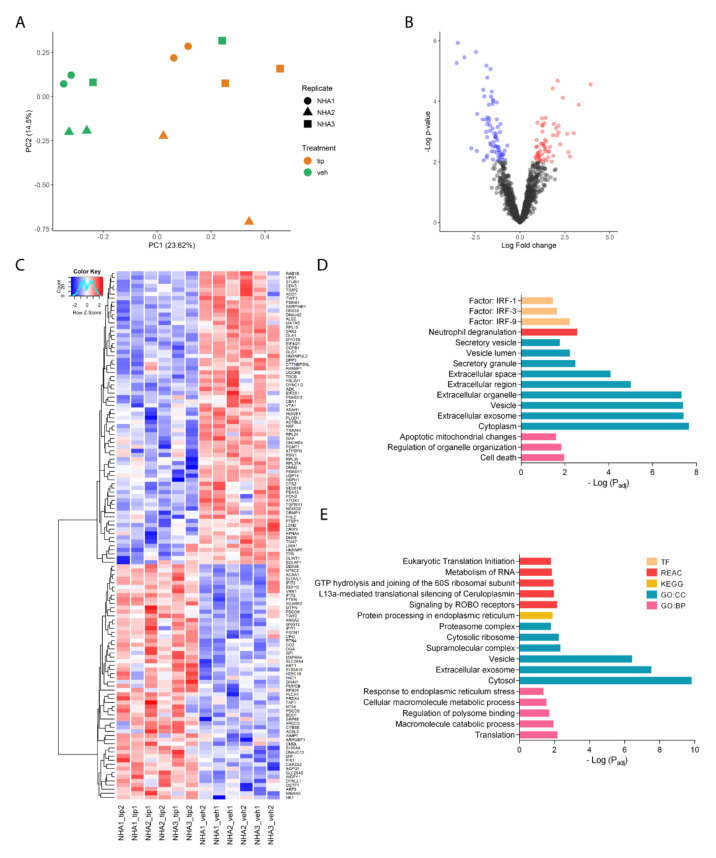
Differential expression of tip vs. veh. (**A**) PCA to evaluate the relationships between biological replicates and tip vs. veh treatments. (**B**) Volcano plot showing differentially expressed proteins with adjusted *p*-value < 0.01 and |Fold change| ≥ 1.5. (**C**) Heatmap of proteins differentially expressed proteins in tip vs. veh. Color scale: variation of normalized protein abundance (Z-Score). (**D**) Enrichment analysis of up-regulated proteins in tip vs. veh (*p* < 0.05). (**E**) Enrichment analysis of down-regulated proteins in tip vs. veh (*p* < 0.05). TF: TRANSFAC database; REAC: Reactome database; KEGG: Kyoto Encyclopedia of Genes and Genomes; GO:CC: Gene Ontology cellular component; GO: BP: Gene Ontology biological process.

**Figure 4 ijms-23-06454-f004:**
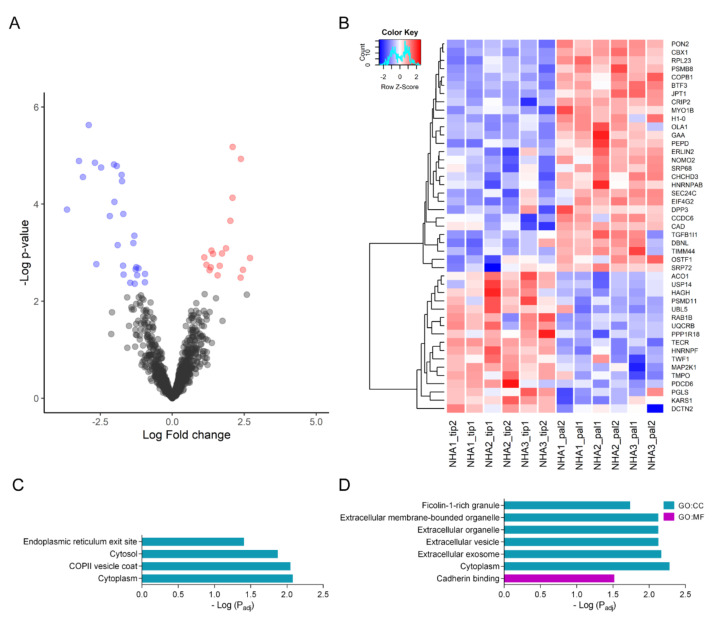
Differential expression of tip vs. pal. (**A**) Volcano plot showing differentially expressed proteins with adjusted *p*-value < 0.01 and |Fold change| ≥ 1.5. (**B**) Heatmap of proteins differentially expressed proteins in tip vs. pal. Color scale: variation of normalized protein abundance (Z-Score). (**C**) Enrichment analysis of up-regulated proteins in tip vs. pal (*p* < 0.05). (**D**) Enrichment analysis of down-regulated proteins in tip vs. pal (*p* < 0.05). GO:CC: Gene Ontology cellular component; GO:MF: Gene Ontology molecular function.

**Figure 5 ijms-23-06454-f005:**
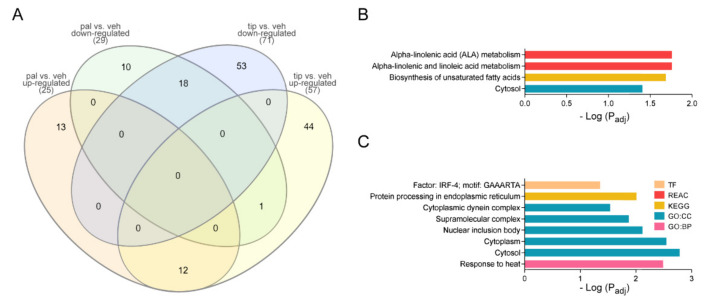
Analysis of the intersect proteins in pal vs. veh and tip vs. veh. (**A**) Venn diagram of pal and tip vs. veh. In the diagram, tip vs. veh has 44 up-regulated unique proteins and 53 down-regulated unique proteins; pal vs. veh has 13 up-regulated unique proteins and 10 down-regulated unique proteins; between the comparisons, there are 12 up-regulated y 18 down-regulated shared. (**B**) Functional enrichment of shared up-regulated proteins. (**C**) Functional enrichment of shared down-regulated proteins.

**Figure 6 ijms-23-06454-f006:**
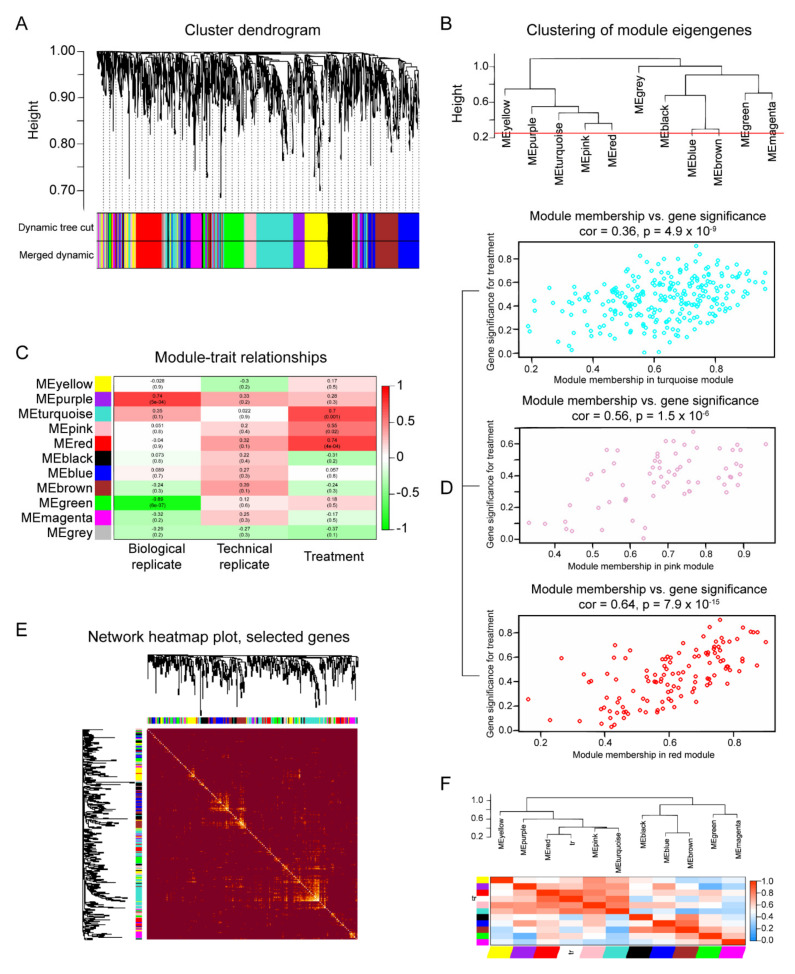
WGCNA analysis. (**A**) Average linkage hierarchical clustering gene dendrogram showing modules identified by Dynamic Tree Cut and merged dynamic at the lower side. (**B**) Dendrogram of module eigengenes (ME) identified by their colors, displaying the modules found in the clustering analysis, the red line represents the cutting-edge for merging modules. (**C**) Heatmap plot displaying the correlation of the adjacencies in the eigengene modules (rows) along treatment and biological and technical replicates. (**D**) Scatterplots of gene significance (GS) versus module membership (MM) for the modules “red”, “pink”, and “turquoise”. (**E**) Topological overlap in the gene network, brighter squares show modules of expression displaying high topological overlap, darker areas represent low topological overlap between proteins in the network (rows and columns). (**F**) Heatmap of the correlation between different modules and the weight trait (tr).

**Figure 7 ijms-23-06454-f007:**
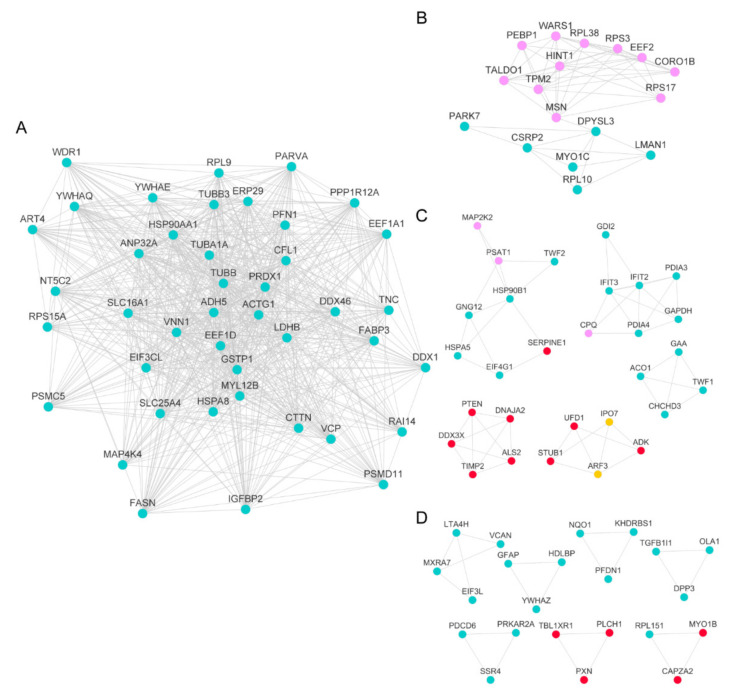
Protein-protein interaction network of the highly correlated modules of proteins from WGCNA. Hub-proteins in the red, pink, and turquoise modules after using the MCODE algorithm of Cytoscape to reduce the nodes for those with more weight in the network from the whole data set. (**A**) Cluster 1 with an MCODE score of 35,220 and 42 nodes. (**B**) Cluster 2 with a score of 7375 and 17 nodes. (**C**) Clusters 3 to 7 with MCODE score >3500. (**D**) Clusters 8 to 14 with MCODE score >3000. Node colors correspond to their original modules, except for IPO7 and ARF3, shown in yellow.

**Figure 8 ijms-23-06454-f008:**
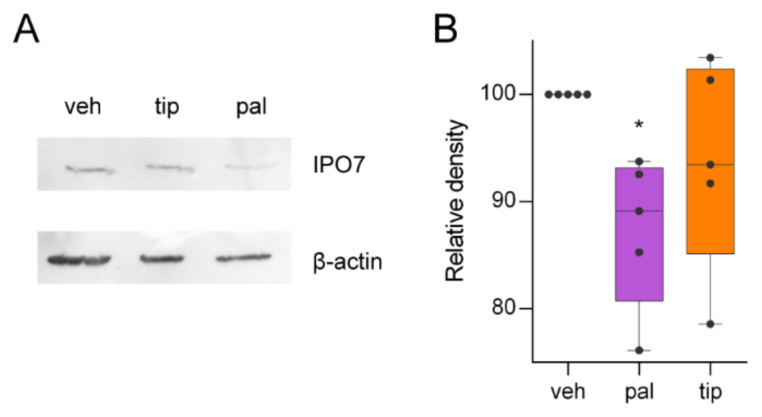
Western blot validation of the IPO7 expression. (**A**) Representative western blot of IPO7 expression under the three different treatments (load control: β-actin). (**B**) Statistic analysis of the band intensity from the western blot of IPO7 normalized by the control band signal (veh). One-way ANOVA and a Tukey’s test of veh vs. pal were used for comparisons; *: *p* = 0.036.

**Table 1 ijms-23-06454-t001:** Shared proteins of treatments compared with the control.

Shared Down	Shared Up
ADK	ACAA1
ALS2	AIMP1
ATP5PO	ARHGEF1
CLINT1	CMBL
COPB1	ELOVL1
CTTNBP2NL	GNG12
DDX3X	IFIT2
DNAJA2	MT-CO2
DYNC1I2	PDCD5
EIF2S1	PTEN
LSM2	RPS25
MYO1B	XRCC5
RPL15	
RSU1	
SRP68	
STUB1	
TIMP2	
TPR	SRP68 (up-regulated in tip vs. veh)
UFD1	

**Table 2 ijms-23-06454-t002:** Key proteins. List of hub proteins that were found in the differentially expressed proteins in each comparison.

Upregulated			Downregulated		
pal vs. veh	tip vs. veh	tip vs. pal	pal vs. veh	tip vs. veh	tip vs. pal
ARF3	MAP4K4 HGK KIAA0687 NIK	None	ALS2 ALS2CR6 KIAA1563	ALS2 ALS2CR6 KIAA1563	CHCHD3 MIC19 MINOS3
PTEN MMAC1 TEP1	PLCH1 KIAA1069 PLCL3		STUB1 CHIP PP1131	TGFB1I1 ARA55	TGFB1I1 ARA55
	PDCD6 ALG2		ADK	CHCHD3 MIC19 MINOS3	ACO1 IREB1
	CAPZA2		DNAJA2 CPR3 HIRIP4	MYO1B	TWF1 PTK9
	TWF2 PTK9L MSTP011		TIMP2	STUB1 CHIP PP1131	DPP3
	PTEN MMAC1 TEP1		IPO7 RANBP7	EIF4G1 EIF4F EIF4G EIF4GI	OLA1 GTPBP9 PRO2455 PTD004
	NT5C2 NT5B NT5CP PNT5		UFD1 UFD1L	TIMP2	GAA
	CPQ LCH1 PGCP		MYO1B	DNAJA2 CPR3 HIRIP4	
	DDX46 KIAA0801			ADK	
	IFIT3 CIG-49 IFI60 IFIT4 ISG60			TWF1 PTK9	
	VNN1			DDX3X DBX DDX3	
	GNG12			OLA1 GTPBP9 PRO2455 PTD004	
	SLC25A4 ANT1			DPP3	
	IFIT2 CIG-42 G10P2 IFI54 ISG54			RPL15 EC45 TCBAP0781	
				UFD1 UFD1L	
				PSMD11	
				SERPINE1 PAI1 PLANH1	

## Data Availability

The data presented in this study are available in the present article and Appendix A.

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
