# Peer review of "Tibolone Pre-Treatment Ameliorates the Dysregulation of Protein Translation and Transport Generated by Palmitic Acid-Induced Lipotoxicity in Human Astrocytes: A Label-Free MS-Based Proteomics and Network Analysis"

_ijms, 2022, doi:10.3390/ijms23126454_

Round 1

Reviewer 1 Report

Manuscript proposed by Vesga-Jimenez and co-workers (ijms-1726735) entitled “Tibolone ameliorates the dysregulation of protein translation and transport generated by palmitic acid-induced lipotoxicity in human astrocytes: a label-free MS-based Proteomics and Network Analysis” presents the application of a proteomic approach to normal human astrocytes under supraphysiological levels of palmitic acid as a model to induce cytotoxicity. The authors found regulation in protein related to translation, transport, autophagy, and apoptosis. Additionally, the role of tibolone as a protective agent was presented. In my opinion, the manuscript present great work however, before publication, needs a revision. 

My major comments are presented below.

Major concerns:

- Abstract – clearly present the novelty of the proposed method in such kind of study.

- Introduction – present paragraph showing the advantages of proteomics approach in the presented topic

- Result section, page 3, Figure 1 – low quality of the figure

- Result section, page 3, lines 122-126 – correct the notation - Result section, page 3, lines 122-126 – correct the notation – is  (p_adj = 1.49 × 10−2) and biosynthesis of unsaturated FAs (p_adj = 1.55 × 10−3) – should be - (p_adj = 1.49 × 10−2) and biosynthesis of unsaturated FAs (p_adj = 1.55 × 10−3). The same on pages 4, 5, 6, 7, 12, 13, 14, 15.

- Result section, page 4, Figure 2 – low quality of the figure

- Materials and methods section – line 497, correct the subscripts. The same on page 17, line 523.  

- 4.5. Protein digestion and load in the Q exactive – page 17 – correct Lys-c to Lys-C

- 4.5. Protein digestion and load in the Q exactive – page 17  - was the collision energy in MS/MS experiment optimized?

- 4.5. Protein digestion and load in the Q exactive – page 17, line 546  - Peptides were eluted using a 180-minute gradient with a 300 nL/min flow rate – what was the gradient? Was the gradient condition optimized?

- 4.5. Protein digestion and load in the Q exactive – page 17 – what was the injection volume?

- 4.5. Protein digestion and load in the Q exactive – page 17 – determine the ion source parameter, analysis mode (positive, negative).

- Materials and method section – lack of information about the purity of used chemicals (solvents for LC-MS/MS analysis, water, etc.)

- how many times was one sample analyzed?

Make changes in the text.

Check and correct English

Author Response

Response to Reviewer 1 Comments

We are very grateful with the reviewer for taking the time to read our manuscript and for your comments and corrections, we followed all your indications, correcting the major concerns pointed out, or answered the questions made, and we trust the quality of the paper has improved after the revision.

- Abstract – clearly present the novelty of the proposed method in such kind of study.

Response: We modified the abstract presenting the novelty and advantages of the method used in our study

- Introduction – present paragraph showing the advantages of proteomics approach in the presented topic

 Response: We introduced a paragraph from the line 87 to 93 explaining the advantages of proteomics approach for the present topic

- Result section, page 3, Figure 1 – low quality of the figure

 Response: We changed the Figure 1 for another one with better resolution

- Result section, page 3, lines 122-126 – correct the notation - Result section, page 3, lines 122-126 – correct the notation – is (p_adj = 1.49 × 10−2) and biosynthesis of unsaturated FAs (p_adj = 1.55 × 10−3) – should be - (p_adj = 1.49 × 10−2) and biosynthesis of unsaturated FAs (p_adj = 1.55 × 10−3). The same on pages 4, 5, 6, 7, 12, 13, 14, 15.

 Response: We corrected the notation in the whole manuscript as suggested

- Result section, page 4, Figure 2 – low quality of the figure

 Response: We changed the Figure 2 for another one with better resolution

- Materials and methods section – line 497, correct the subscripts. The same on page 17, line 523.

Response: Subscripts were corrected on line 521 and line 551

- 4.5. Protein digestion and load in the Q exactive – page 17 – correct Lys-c to Lys-C

 Response: The change was executed

- 4.5. Protein digestion and load in the Q exactive – page 17  - was the collision energy in MS/MS experiment optimized?

Response: Following the specifications of the service hired with UC davis, the collision energy is called normalized collision energy (NCE) which means that the energy is essentially optimized to work over a wide range of analytes in this case peptides.

- 4.5. Protein digestion and load in the Q exactive – page 17, line 546  - Peptides were eluted using a 180-minute gradient with a 300 nL/min flow rate – what was the gradient? 

 Response: The gradient was 140-minute separation and 180-minute total with the loading. Doing it on the following way, A solvent is water with 0.1% formic acid B solvent is Acetonitrile with 0.1% formic acid.

0-92 minutes 5-20%B, 92-112 minutes, 20%B to 32%B, 112-119 minutes 32%B to 80%B, 119-129 minutes hold at 80%B and then at 130 minutes B solvent back down to 5% to equilibrate for 10 minutes, finishing the run at 140 min.

Was the gradient condition optimized? 

Response: Yes, we adjust the gradient often for the changing column and trap conditions to get the best separation over the gradient length which we determine by running a HELA digestion standard.

- 4.5. Protein digestion and load in the Q exactive – page 17 – what was the injection volume? 

 Response: The injection volume was 10µl

- 4.5. Protein digestion and load in the Q exactive – page 17 – determine the ion source parameter, analysis mode (positive, negative). 

Response: The spray voltage was 2.0kV, Capillary temp is 250c, not gas was used, also peptides are run in positive

 - Materials and method section – lack of information about the purity of used chemicals (solvents for LC-MS/MS analysis, water, etc.) 

 Response: We asked to the proteomics service providers, and they answered that they used Fisher Optma LC-MS/MS grade chemicals,

- how many times was one sample analyzed?

Response: Samples were analyzed once

We hope that our manuscript is now suitable for publication in International Journal of Molecular Sciences.

Sincerely,

Janneth González

Reviewer 2 Report

Ref: ijms-1726735

Title: Tibolone ameliorates the dysregulation of protein translation and transport generated by palmitic acid-induced lipotoxicity in human astrocytes: a label-free MS-based Proteomics and Network Analysis

Recommendation: Reject

Overall

The aim of the present paper was to find the actions of tibolone upon lipotoxic damage in human astrocytes. To verify the hypotheses, the authors performed MS-based proteomics and network analysis to assess the effects palmitic acid and tibolone pre-treatment on human astrocytes. The authors conclude that pre-treatment with tibolone possess protective effects against palmitic acid toxicity with ARF3 and IPO7 proteins as key factors.

This paper has a low translational value due to the pre-treatment model used. Moreover, the presented results lack of an important research group which would be the vehicle + tibolone.

Comments:

  1. Low translational value due to the pre-treatment model used.
  2. Lack of an important research group which would be the vehicle + tibolone.
  3. The title should contain ‘pre-treatment’ word. It is important issue for future readers.
  4. The research is based solely on proteomics, there is no testing of protein action/function, and the up- / down-regulation itself not always act 1:1 with the function.
  5. The authors do not mention much about the uniqueness of tibolone.
  6. The used concentrations of palmitic acid and tibolone are mentioned only in Materials and Methods section.
  7. Section 4.1. should be described in great detail to allow the experiment to be repeated by others.
  8. Why there is 6-h serum deprivation?

Author Response

We want to thank the reviewer for taking the time to read and suggest corrections for our manuscript, we have addressed all the suggestions made by the reviewer making considerable modifications to the manuscript, focusing on improving the methodology description to make it clearer and easier to read, and correction of different minor issues. We believe that after the round of revision the quality of the manuscript Improved considerably.

Comments:

  1. Low translational value due to the pre-treatment model used.

Response:

We agree with the reviewer. It is highly desirable that these types of studies achieve translational power; however, the process of translation could be long, complex, and expensive. The novelty and value of this research are described in detail in the following answers.

Lack of an important research group which would be the vehicle + tibolone.

Response: We acknowledged that in this study we did not include the research group vehicle + tibolone for the following reasons: Previously in a study published by our group (Martin-Jimenez et al., 2020) we reported tibolone at 10nM protects astrocytes by improving cellular survival, and this is correlated to the preservation of mitochondrial function and integrity when exposed to palmitic acid.  There were no significant changes between tibolone vs vehicle-treated astrocytes. To confirm these previous results, as a pilot study prior to proteome assessment few samples comparing the DMEM alone and tibolone + DMEM showed very small differences compared to the other groups. Given this, as to avoid a significant increase in costs and time-consuming experiments for proteome isolation and analysis we decided not to include this group (vehicle + tibolone) in the current study based on our previous results. More importantly is the focus of the current paper to delve into the mechanisms of tibolone protection against palmitic acid that hadn’t been explored previously,  and not the changes induced by tibolone pre-treatment because the changes of estrogenic responses with tibolone and estrogen have been already reported (Laakonen et al., 2017).

Martin-Jiménez C, González J, Vesga D, Aristizabal A, Barreto GE. Tibolone Ameliorates the Lipotoxic Effect of Palmitic Acid in Normal Human Astrocytes. Neurotox Res. 2020 Oct;38(3):585-595. doi: 10.1007/s12640-020-00247-4. Epub 2020 Jul 7. PMID: 32638213.

Laakkonen, E. K., Soliymani, R., Karvinen, S., Kaprio, J., Kujala, U. M., Baumann, M., Sipilä, S., Kovanen, V., & Lalowski, M. (2017). Estrogenic regulation of skeletal muscle proteome: a study of premenopausal women and postmenopausal MZ cotwins discordant for hormonal therapy. Aging cell, 16(6), 1276–1287. https://doi.org/10.1111/acel.12661

The title should contain ‘pre-treatment’ word. It is important issue for future readers.

Response: We modified the title including pre-treatment line 1

The research is based solely on proteomics, there is no testing of protein action/function, and the up- / down-regulation itself not always act 1:1 with the function.

Response: We agree with the reviewer. This is one of our next future goals, that is a complete functional characterization of IPO7 in astrocytes. It is important to emphasize that the present study is the first to investigate how tibolone exerts its neuroprotective effects when astrocytes are stimulated with palmitic acid. The novelty of the current paper is precisely this, since so far it has not been satisfactorily explored. On the other hand, as per the reviewer’s comment, it is essential to explore the biological function/action of the protein and if regulating its expression (up/down) what would be the final biological response when cells are in a lipotoxic environment. We are currently conducting these studies using CRISPR to inhibit or overexpress IPO7, in the presence or absence of tibolone, to determine whether this protein is important in mediating, at least in part, the anti-inflammatory functions of tibolone. We will soon be reporting these results in a new article.

 The authors do not mention much about the uniqueness of tibolone.

Response: We added a more detailed explanation of tibolone relevance in the introduction line 75 and compared the results obtained by us and another proteomics study that used tibolone line 416

The used concentrations of palmitic acid and tibolone are mentioned only in the Materials and Methods section.

Response: We added the concentrations on other parts of the text line 120, 12, 136, 160, 189, 297, 327, 332, 342, 409, 435, 471, 659, 660

Section 4.1. should be described in great detail to allow the experiment to be repeated by others.

Response: We corrected and explained this section in a more detailed way in order that the experiments can be carried out easier by other researchers

Why there is 6-h serum deprivation?

Response: we perform serum starvation to reduce analytical interference, reduce the basal activity of cells, and provide more reproducible experimental conditions. Additionally, we tested different starvation times to ensure that astrocytes did not develop deleterious responses

  • Colzani M , Waridel P , Laurent J , Faes E , Ruegg C , Quadroni M. Metabolic labeling and protein linearization technology allow the study of proteins secreted by cultured cells in serum-containing media. J Proteome Res 8: 4779–4788, 2009.
  • Lambert K , Pirt SJ. Growth of human diploid cells (strain MRC-5) in defined medium; replacement of serum by a fraction of serum ultrafiltrate. J Cell Sci 35: 381–392, 1979.
  • Mbeunkui F , Fodstad O , Pannell LK. Secretory protein enrichment and analysis: an optimized approach applied on cancer cell lines using 2D LC-MS/MS. J Proteome Res 5: 899–906, 2006.

We hope that our manuscript is now suitable for publication in the International Journal of Molecular Sciences.

Sincerely,

Janneth González

Round 2

Reviewer 2 Report

The authors made significant changes to the manuscript. They also answered all my questions / comments. Currently, in my opinion, the manuscript is ready for publication.